# Environmental Sensing in High-Altitude Mountain Ecosystems Powered by Sedimentary Microbial Fuel Cells

**DOI:** 10.3390/s23042101

**Published:** 2023-02-13

**Authors:** Celso Recalde, Denys López, Diana Aguay, Víctor J. García

**Affiliations:** 1Facultad de Ciencias, Escuela Superior Politécnica de Chimborazo, Riobamba EC060155, Ecuador; 2Facultad de Ingeniería, Carrera de Ingeniería Civil, Universidad Nacional de Chimborazo, Riobamba EC060108, Ecuador

**Keywords:** maximum power point tracking, unconventional sensing, dynamic environmental sensing, local sensing environmental variables, distributed and local sensing, wireless sensor network, energy harvesting, environmental water monitoring, anoxic environmental conditions

## Abstract

The increasing need for fresh water in a climate change scenario requires remote monitoring of water bodies in high-altitude mountain areas. This study aimed to explore the feasibility of SMFC operation in the presence of low dissolved oxygen concentrations for remote, on-site monitoring of physical environmental parameters in high-altitude mountainous areas. The implemented power management system (PMS) uses a reference SMFC (SMFC_Ref_) to implement a quasi-maximum power point tracking (quasi-MPPT) algorithm to harvest energy stably. As a result, while transmitting in a point-to-point wireless sensor network topology, the system achieves an overall efficiency of 59.6%. Furthermore, the control mechanisms prevent energy waste and maintain a stable voltage despite the microbial fuel cell (MFC)’s high impedance, low time response, and low energy production. Moreover, our system enables a fundamental understanding of environmental systems and their resilience of adaptation strategies by being a low-cost, ecological, and environmentally friendly alternative to power-distributed and dynamic environmental sensing networks in high-altitude mountain ecosystems with anoxic environmental conditions.

## 1. Introduction

A fundamental understanding of environmental systems, their resilience, and adaptation strategies drive innovative technology development in distributed and dynamic environmental sensing via novel methodologies/technologies (unconventional sensing involving living organisms). Therefore, environmental monitoring has become a global concern [1,2]. For example, the growing need for fresh water requires remote monitoring of water bodies in mountain areas. In addition, microelectronic advances enable a wide range of sensors in remote locations for monitoring habitat, climate, agricultural applications, fisheries research, health systems, air, soil, and water [1,3]. Energy sources of the monitoring systems must be renewable and reliable and not require human intervention [2]. The main concern of implementing a distributed and local sensing of environmental variables via novel methodologies/technologies is the reliability of primary energy sources. Often energy harvesting depends on environmental conditions, and energy can be available when the primary source is available. Energy harvesting or scavenging technology has recently gained attention for clean and sustainable energy [4]. Thus, some alternative energy sources depend on physical conditions, such as the availability of light, wind, heat, and others [2].

Environmental variables sensing and monitoring systems in remote areas use mainly collected energy. It is often not logistically feasible or sustainable due to the high implementation and maintenance costs. Moreover, the monitoring system requires capacitors or disposable batteries, which contain toxic, environmentally hazardous materials, so their use for water monitoring is little appreciated [5]. An alternative energy source for sensing and monitoring systems for water bodies is SMFC [2]. The SMFC generates electricity from sediments rich in organic matter buried in anoxic conditions (depleted of dissolved oxygen) [6,7,8,9,10,11]. SMFC competes with biomass combustion and anaerobic digestion [12]. The SMFC does not contain separators or membranes. Instead, the cathode electrodes accept electrons and protons to reduce oxygen to water, taking advantage of the dissolved oxygen gradient of water concerning depth [13]. Hence, the feasibility of SMFC operation under low dissolved oxygen concentration becomes a significant study in high-altitude mountain areas [14].

Moreover, SMFCs are very convenient for monitoring devices of water bodies in high altitude mountain ecosystems with Andosol soils rich in organic matter, low apparent density, and over 4000 masl [15]. These ecosystems are huge freshwater reservoirs [15,16]. Meybeck et al. [17] estimated that 32% of the planet’s freshwater is on high-altitude mountains. Under these conditions, the oxygen availability is around one-fifth of that at sea level, and its solubility is substantially reduced [15].

Although SMFC produces little energy, tests conducted in places close to sea level have shown the SMFC potential as an energy source for monitoring systems in remote areas. When an electric load is applied to stack MFCs, the stack MFC’s voltage does not remain constant over time, known as the reversal of voltage. Therefore, the actual direction is focused on the design of a PMS adapted for MFC optimal operation using ultralow power consumption components to raise the voltage of the MFC [6]. For example, SMFCs powered a sensor installed in the Palouse River [18,19]. SMFC with sediments extracted from Lake Michigan intermittently powered a temperature sensor [20]. Many reports have shown the SMFC’s potential to power environmental wireless sensor networks [21]. The main point is to use the appropriate electronic circuit to transfer and store the electric energy produced by the SMFCs [20,22,23] PMS was used to operate remote sensors powered by SMFCs [24,25]. The PMS was used to control energy harvesting, store energy, and channel the necessary power for data transmission [2,6,7,21]. The PMS converts low input to a high output voltage by providing enough power bursts to intermittently control typical commercial electronic devices [13,18,20,21,26,27,28]. However, few studies addressed the feasibility of SMFCs in high-altitude mountain ecosystems with low oxygen solubility [29,30]. 

This study aimed to explore the feasibility of SMFC operation in the presence of low dissolved oxygen concentrations for remote, on-site monitoring of physical environmental parameters in high-altitude mountainous areas. The PMS architecture uses a reference SMFC_Ref_ to implement a quasi-MPPT algorithm to harvest energy effectively.

## 2. Materials and Methods

A wireless sensor network was installed in the laboratory under the IEEE 802.15.4 standard with a point-to-point topology. Four SMFCs power the wireless sensors. A fifth SMFC labeled SMFC_REF_ works as a voltage reference in the charging algorithm for transmitting temperature data to a base station. 

### 2.1. Sedimentary Microbial Fuel Cells

Sediment and water were collected from the Colta lagoon, coordinates −1.725873, −78.757530 at 3212 masl. Five SMFCs were implemented using cylindrical plastic containers with dimensions of 145 mm radius and 200 mm height (Figure 1a) and filled with a 60 mm thick sediment layer. The water is 100 mm above the sediment, and the cathode electrode is installed 40 mm below the water surface. Without any treatment, the anode and cathode electrodes were made of carbon fiber cloth whose dimensions were 100 mm × 100 mm and 100 mm × 150 mm, respectively. The electrodes were attached with a 100 mm × 20 mm stainless steel mesh and connected to the outside with a 100 mm long nickel/titanium alloy wire. The anode electrode was installed 40 mm inside the substrate and connected outside the container with a cork (Figure 1).

Each SMFC was connected to a 100 Ω resistance to improve the establishment of the biofilm at room temperature without adding additional nutrients during the 30 days. In addition, water collected from the lagoon was poured periodically to compensate for water loss due to evaporation.

The differential voltages of individual SMFC in steady-state were monitored every minute for 48 h. Subsequently, four SMFCs were connected in parallel (Figure 1b) to increase the power generated. Data were collected in both cases with a NI DAQ SB-6009 card (National Instruments Corporation, Austin, TX 78759, USA) and a LabVIEW virtual instrument. The polarization and power curves of the SMFCs in the parallel circuit were obtained with the resistance box method using variable precision resistors of 2 MΩ, 10 kΩ, and 1 kΩ. The SMFC’s current was calculated using Ohm’s law (V=IR) and the power equation (P=VI); these parameters were used to graph polarization and power curves. Finally, the fifth SMFC was connected to a variable resistor to obtain a percentage of its open-circuit voltage (OCV) and used in the quasi-MPPT algorithm.

### 2.2. Power Management System

The PMS uses a bq25505 and the tps61200 IC. The bps25505 IC is from Texas Instruments (Dallas, TX, USA), an ultra-low-power battery lifter and charger specialized in high-impedance low-power sources. The tps61200 from Texas Instruments (Dallas, TX, USA) is a high-efficiency synchronous DC–DC boost converter with a configurable voltage output from 1.8 V to 5.5 V against over-voltage and current. Figure 2 shows the implemented system where the bq25505 extracts the charges from the SMFCs, allowing input voltages as low as 100 mV. The charges are stored in a 0.7 μF supercapacitor (SC). When the SC reaches 2.44 V (VBAT_OK_PROG), it activates a control signal (VBAT_OK). The signal deactivates when it falls below 2.38 V (VBAT_OK_HYST). The VBAT_OK_PROG, VBAT_OK_HYST, and other supercapacitor protection parameters are configured using Rok1, Rok2, Rok3, and Rov1, 2 external resistors (Figure 2a). 

The maximum power point (MPP) observed in SMFCs is usually around 50% of their open circuit voltage (OCV). However, when the MPPT module (bq25505) was set to 50% of the OCV, the circuit operation stopped after a few minutes and started working. This phenomenon was attributed to the slow voltage recovery of the SMFCs in open-circuit. The phenomenon was reported by Degrenne et al. [31], and Alaraj and Park [32] used an MFC model’s perturbation and observation algorithm to solve this issue. These authors estimated its MPP to modulate the behavior of a bq25504 to obtain an energy harvest close to real MPP with fast convergence.

The SMFC_Ref_ was connected to a variable resistor whose output was attached to the VREF_SAMP terminal that controls the voltage at which the MPPT works (Figure 2a). The variable resistor was calibrated to obtain at its output 50% of the OCV and to be similar to the OCV of the SMFCs with parallel connection. This process avoids estimating the MPP through the direct observation of the OCV or some parameter of the SMFCs. Instead, similar behavior of the SMFC_Ref_ and the SMFCs operating under identical conditions is assumed [33,34].

We used the multi-unit optimization method algorithm [35], with correction parameters calculated for non-identical units, to determine the MPP of an MFC with relatively slow dynamics.

The SMFC_Ref_ and the SMFCs have different internal resistances, so their MPPs are not coincident. However, previous reports suggest that their OCV present similar and stable behaviors under the same conditions. Therefore, it is an excellent approximation to determine the MPP using a SMFC_Ref,_ considering a stable and smooth behavior from the MFCs, implementing a PMS with a quasi-MPPT [11,36,37].

Once the supercapacitor reaches the charging voltage of 2.44 V (VBAT_OK_PROG), the tps61200 module regulates and raises the final voltage level to 3.3 V, activated by the VBAT_OK signal connected to the EN pin (Figure 2).

Once the capacitor voltage drops below 2.38 V (VBAT_OK_HYST), the sensor signal (SEN) generated by the load activates an Opto-MOSFET to keep the converter active until the transmission process has been accomplished (Figure 2a).

### 2.3. Load System

Figure 3 shows the load block diagram consisting of a PIC 16f628A microcontroller (Texas Instruments Co., Austin, TX, USA) as a node controller. A ds18b20 sensor (Hefei Jingpu Sensor Technology Co., Hefei, China) was used as a device to acquire temperature values using the 1-wire communication protocol, with a resolution of 9 bits with an error of ±0.5 °C. In addition, a radio frequency module Xbee ZB S2C TH (Digi International, Hopkins, MN, USA) with IEEE 802.15.4 standard support for creating point-to-point networks in transparent mode was used.

The system starts the microcontroller when it receives the 3.3 V power supply (VCC) from the PMS and immediately turns the SEN signal on to keep the PMS output activated (Figure 3). Then it enters standby mode with consumption around 100 nA, waiting for the ASSOC signal (Figure 3) emitted by the radio frequency module, indicating that it is ready to transmit. The ds18b20 takes a temperature value and sends it through the universal asynchronous receiver transmitter (UART) port for transmission. The transmission format follows the JavaScript Object Notation (JSON) key/value pair format, sending the node identifier and the measured temperature value {“Id”: “1”, “T1”: “22.1”} as parameters. A total of 22 bytes are sent during each transmission.

A watchdog timer (WDT) was built to automatically reset the microcontroller when no response to the ASSOC signal was received within 2.3 s. Hence, it disconnects the PMS from the load by setting the SEN signal low and preventing energy waste if the connection with the XbeeRF module is not achieved.

Table 1 shows the power consumption of the devices according to the manufacturer, with the Xbee RF module consuming approximately 92.6% of the total power.

### 2.4. Base Station System

The base station has an Xbee ZB S2C TH (Digi International) radio frequency module to receive the data sent by the sensor node. In addition, it works with a UDOO NEO Mini PC, intercommunicating with its UART interfaces.

### 2.5. System Tests and Calculations

#### 2.5.1. Load Consumption Test

A 6.8 Ω shunt resistor was used to calculate the actual load consumption, whose voltage $V_{shunt}$ was measured every ms with an NI USB-6009 card when the load consumed energy in a temperature data transmission. The current flowing through the load was calculated using Equation (1). The voltage across the load is the difference between the voltage of the power supply and the voltage across the shunt resistor, calculated using Equation (2).

#### 2.5.2. Voltage Capacitor Test

Voltage measurements were performed at the cold start and transmission modes. The capacitor, PMS input, and output voltages were registered by a NI USB-6009 card over twenty hours, as shown in Figure 4. The PMS uses four SMFCs connected in a parallel configuration as an energy source. In contrast, the SMFC_REF_ uses a variable resistor tuned until its output voltage equals the voltage where the maximum power point was found.

The test was conducted to determine the behavior of the input and output voltages of the PMS, and to determine how the voltage of the capacitor changes from when it is empty to when it has enough energy for the sensor node to transmit intermittently.

Equations (3) and (4) calculate supercapacitor power and absorbed energy per unit of time. Equations (5) and (6) calculate the converter’s efficiency. Three tests were conducted to compute the load consumption.
(1)$$I_{load}=\frac{V_{shunt}}{6.8Ω}$$
(2)$$V_{load}=3.3V-V_{shunt}$$
(3)P=E/t
(4)$$E=\frac{1}{2}C\left(V_{2}{}^{2}-V_{1}{}^{2}\right)$$
(5)$$\eta_{bq25505}=\frac{E_{scap}}{E_{MFC}}\times 100$$
(6)$$\eta_{tps61200}=\frac{E_{load}}{E_{scap}}\times 100$$

## 3. Results and Discussion

### 3.1. Sedimentary Microbial Fuel Cells

After passing the biofilm maturation stage, after around 34 days, the average voltages recorded in the four SMFCs in one day were: 0.49, 0.47, 0.53, and 0.41 V, and the average voltage of the four SMFCs in parallel configuration was 0.47 V, similar to those average voltages in oxygen-rich water in ponds at Milan (Italy) [1]. The SMFC_Ref_ reached a 0.45 V average voltage. Figure 5 shows the polarization and power curves obtained from the SMFC_s_ in a parallel configuration.

The polarization curve fitted to the equation $V=-0.254*10^{3}I+0.4748$, a simple linear regression model with a coefficient of determination, $R^{2}$, of 99.8%. The slope of this line is equivalent to the internal resistance $R_{\mathrm{int}}$ mainly due to the ohmic losses, so the internal resistance of the four parallel SMFCs is approximately 254 Ω. The power curve follows a second-degree polynomial regression model $P=-0.2516I^{2}+0.4714I+0.0006$ with a coefficient of determination, $R^{2}$, of 99.38%. The maximum power delivered by the SMFCs is found at the vertex point of the curve where the slope equals zero, generating a maximum power of 0.2214 mW with a voltage and current of 0.236 V and 0.9368 mA, respectively. The input voltage is low compared to those reported by other authors [4,18,38,39,40,41,42,43,44,45,46]. The MPP voltage is used to calibrate the SMFC_Ref_ as the MPPT reference value [6,21,26].

### 3.2. Load Consumption

Figure 6 shows the voltage and current measured (red and orange lines, respectively) when the load performs a single transmission where the energy consumption is 162.02 mJ in a time of 1.73 s. The voltage drop is due to the measurement shunt resistor’s power consumption.

Despite the microcontroller and the sensor consuming lower current (Table 1), the highest current values are recorded when the radio frequency module goes into operation and transmission mode.

Table 2 shows the power and load energy consumption during the three operating cycles registered. It obtained an average energy consumption of 162 mJ and a power of 93.4 mW, close to the 117.6 mW calculated using the datasheets of the manufacturer’s devices.

### 3.3. Power Management System Operating Cycle

Several PMS systems have been proposed with MPPT systems to extract energy from different MFC [6,7] and photovoltaic systems [21,26]. The objective is to overcome the challenges encountered when employing an integrated circuit not designed for the unique characteristics and dynamics of MFCs [32,34]. In this work, we design a PMS that extracts energy close to the MPP with a dynamic external reference implementing a quasi-MPPT.

Figure 7 shows the supercapacitor’s charging cycle starting its charge with the low-efficiency “cold-start” converter integrated with the bq25505. A transition occurs at values greater than 1.8 V (VSTOR_CHGEN). The “Main Boost Charger” mode of the high-efficiency converter is enabled using the MPPT. As shown in Table 3, the charging time of the Main Boost Charger was 4.72 times greater than that of the cold-start converter. The supercapacitor charges from 0 to 2.44 V in 10.7 h.

Figure 7, inset, depicts the supercapacitor’s charging and discharging cycle in the sensor node transmission stages as its voltage oscillates between 2.44 and 2.32 V. The PMS voltage output in each discharge period was 3.3 V. The temperature data transmitted and registered were 22.5, 22.4, 22.5, and 22.8 °C in about 80 min. At the same time, its input voltage remained constant at 0.236 V MPP value. The energy delivered E_MFC_ by the SMFCs is obtained by multiplying the power delivered (0.2214 mW) by the charging time of each period with energy stored in the supercapacitor (E_scap_). Table 4 shows the energy values stored in the supercapacitor made in the four transmissions of the sensor node and the efficiency of the bq25505.

The Xbee radio frequency module is the device with the predominant consumption in the system, with a value of 92.6% of the total, followed by the temperature sensor with 2.81%. However, this requires only 93.75 ms to obtain and transmit temperature data.

Table 5 summarizes the performance of the proposed PMS powered by SMFCs and the direct losses in the devices. The results in Table 5 suggest that the total efficiency can improve by decreasing the losses of the integrated circuits and the storage device’s choice.

The system can start up from any voltage value in the supercapacitor. Additionally, the system disconnects the load once the task has been completed to prevent energy losses. The proposed approach is similar to a configuration harvest energy with a quasi-MPPT. It presents a similar behavior to those reported in the literature, as shown in Table 6. However, it uses a configuration with a dynamic reference taken from a SMFC_Ref_, which makes the configuration even more straightforward.

The proposed PMS system is categorized as a boost converter-type integrated circuit system with quasi-MPPT [6]. The PMS had a maximum efficiency of 71.2% at the best integrated circuit (IC) performances. The IC can operate with 100 mV (when the supercapacitor voltage has reached 1.8 V), with an output of 3.3 V with a maximum current of 1 A.

Discrete electronic components and ICs have been used to establish strategies to boost energy, harvest energy at the maximum power point, and increase the MFC power and behavior [21,28,47]. However, conventional ICs for energy harvesting do not work with high impedance and low time recovery seen in the SMFCs. Nevertheless, our PMS power by SMFCs has shown to be a viable alternative power source while working in unfavorable conditions. Despite the anoxic conditions of the water collected from the Colta Lagoon, using an integrated circuit-based system and quasi-MPPT is among the most efficient devices (Table 6). For the sake of comparison, Table 6 summarizes the primary performance indicator of PMS powered by MFCs from the literature compared with the prototype presented in this paper. It can be seen that the proposed approach showed reasonably good performance.

## 4. Conclusions

This study implemented a point-to-point wireless sensor network powered by SMFCs to intermittently transmit ambient temperature data to a base station. Our results show the feasibility of SMFC operation in the presence of low dissolved oxygen concentrations for remote, on-site monitoring of physical environmental parameters in high-altitude mountainous areas, although the production of electric current in SMFCs can be affected in the high-altitude mountain where oxygen dissolved in water bodies is usually low.

The implemented PMS continuously and effectively harvests energy from SMFCs. The PMS, the implemented quasi-MPPT, and a reference SMFC enable the extraction of energy stably, reaching an overall efficiency of 59.6%. Many factors affect the MFC’s behavior: temperature, bacterial community, dissolved oxygen in water, and physical parameters of the surrounding environment. The control mechanisms prevent energy waste and maintain a stable voltage despite the MFC’s high impedance, low time response, and low energy production.

Our system is low price, easy to implement, and adapted to loads with different low-power demands. Moreover, our system enables a fundamental understanding of environmental systems and their resilience of adaptation strategies by being an alternative to power-distributed and dynamic environmental sensing networks in high-altitude mountain ecosystems with anoxic environmental conditions. However, more studies are required to investigate the behavior of the system in long-term operation, field applications, and other SMFC architectures.

## Figures and Tables

**Figure 1 sensors-23-02101-f001:**
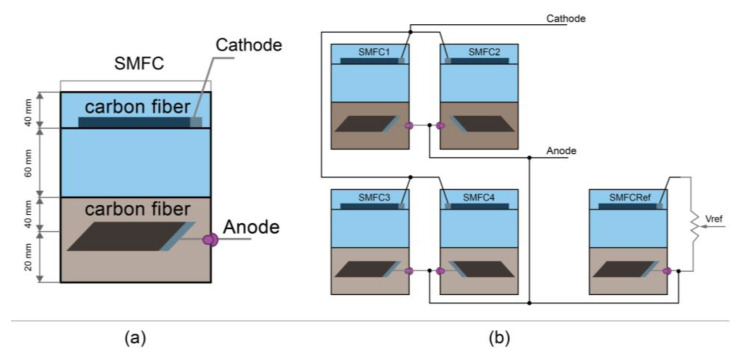
(**a**) Schematic sediment microbial fuel cell diagram. (**b**) Topology of the SMFCs arrangement: four SMFCs connected in parallel and the SMFC_Ref_.

**Figure 2 sensors-23-02101-f002:**
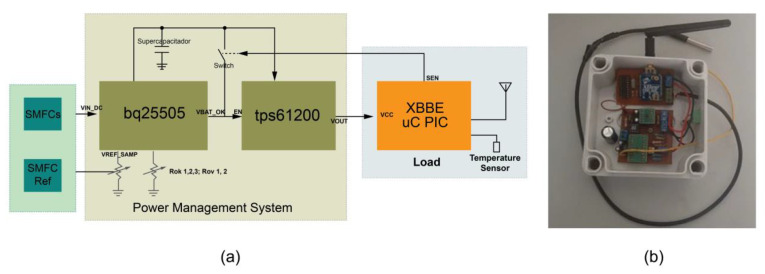
(**a**) Sensor node block diagram. (**b**) The implemented sensor node.

**Figure 3 sensors-23-02101-f003:**
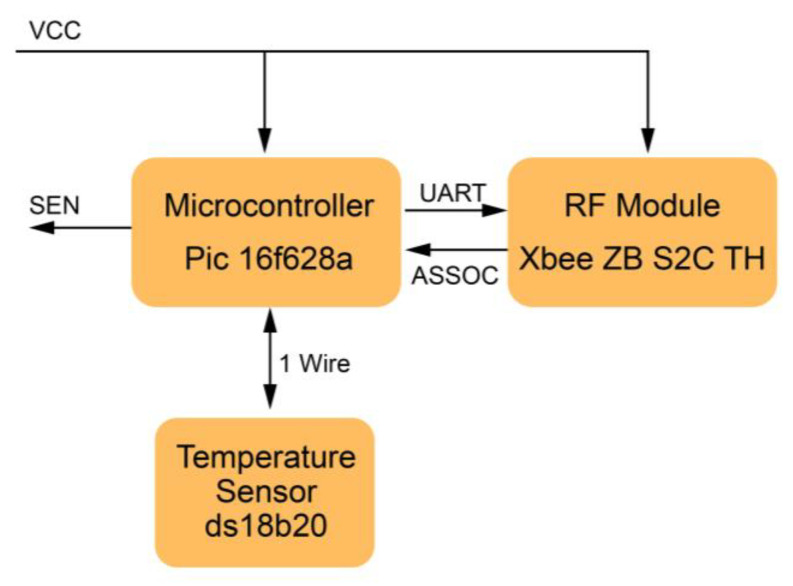
Load block diagram.

**Figure 4 sensors-23-02101-f004:**
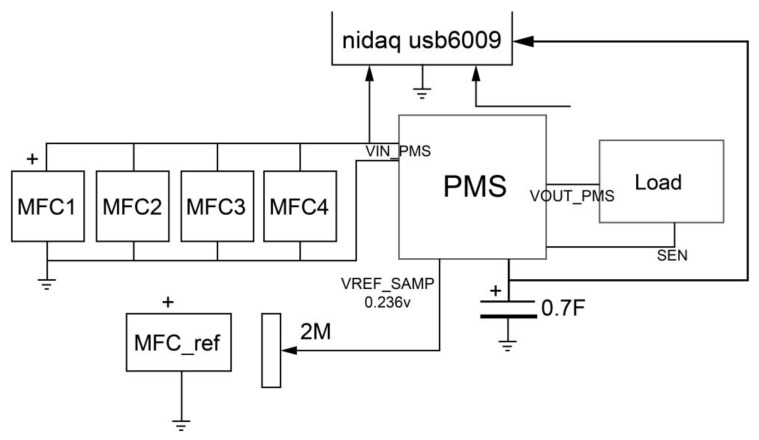
Capacitor and PMS input and output voltage test.

**Figure 5 sensors-23-02101-f005:**
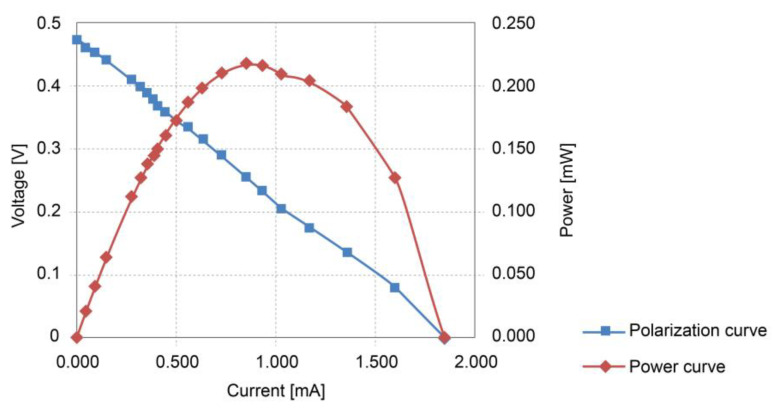
Parallel SMFCs power and polarization curves.

**Figure 6 sensors-23-02101-f006:**
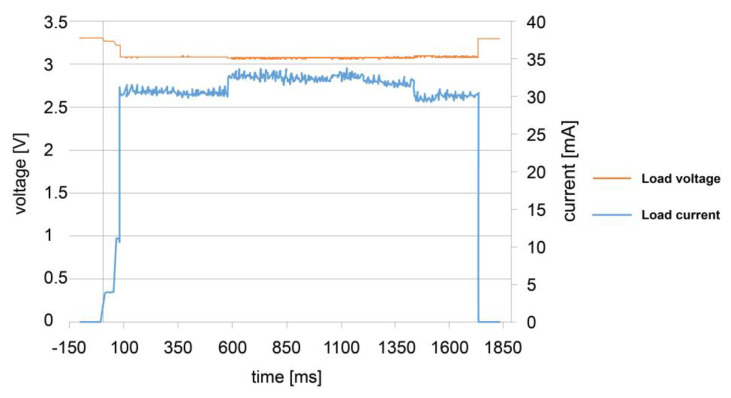
Load voltage and current consumption.

**Figure 7 sensors-23-02101-f007:**
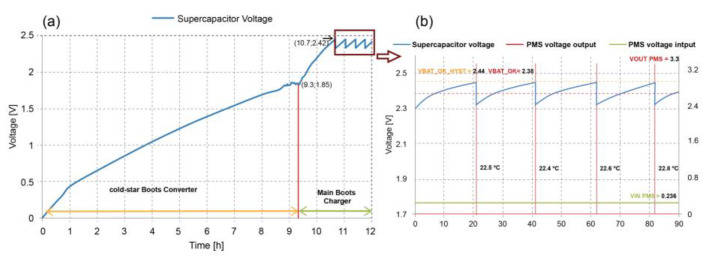
(**a**) Supercapacitor voltage behavior from cold start mode to transmission mode and temperature measured by the system. (**b**) Inset magnification of the supercapacitor’s charging and discharging cycle in the sensor node transmission stages.

**Table 1 sensors-23-02101-t001:** Devices’ current and power consumptions.

Device	Current, mA	Power, mW
Xbee transmission mode	33.00	108.90
PIC 16f628a operative mode 4 MHZ 3.3 V	0.60	1.98
ds18b20 sensor	1.00	3.30
Resistors	1.03	3.40
TOTAL	35.63	117.58

**Table 2 sensors-23-02101-t002:** Values of energy and power in the load.

Transmission	1	2	3	Average
Average power, mW	93.65	93.22	93.28	93.38
Time, s	1.73	1.74	1.72	1.73
Energy, mJ	162.02	162.20	160.91	161.71

**Table 3 sensors-23-02101-t003:** Values of energy stored in the supercapacitor and power supplied by the converters.

Converter	Cold-Start Boost Converter	Main Boost Charger
*V*_1_, V	0.000	1.850
*V*_2_, V	1.850	2.420
Time, h	9.300	1.400
*E_scap_*, J	1.200	0.850
Power, mW	0.036	0.169

**Table 4 sensors-23-02101-t004:** Values of energy stored in each transmission period of the sensor node.

Transmission	1	2	3	4	Average
*V*_1_, V	2.321	2.321	2.325	2.330	2.320
*V*_2_, V	2.438	2.443	2.448	2.443	2.440
Time, min	19.900	20.530	21.100	20.200	20.430
$E_{scap},$ mJ	194.880	203.420	205.480	188.770	198.140
$E_{MFC},$ mJ	264.350	272.720	280.290	268.340	271.430
Efficiency, η_bq25505 (%), Equation (5)	73.720	74.590	73.310	70.350	73.000

**Table 5 sensors-23-02101-t005:** Efficiencies and losses in the PMS.

Output voltage, V	3.300
Input voltage, V	0.236
Input power, mW	0.221
Useful output power, mW	162
Total efficiency, %	59.6
Efficiency tps61200, % (Equation (6))	81.6
Efficiency bq25505, % (Equation (5))	73.0
Loses in tps61200, %	13.4
Loses in bq25505, %	27.0

**Table 6 sensors-23-02101-t006:** Reported primary performance indicator of PMS powered by MFC.

PMS Topology	MFC Type	Voltage Input, V	Efficiency, %	Reference
Capacitor-boost converter	sedimentary	0.400	55	[38]
Capacitor-transformer-boost converter	single chamber	0.475	58	[39]
Capacitor-charge pump-boost converter	sedimentary	0.052–0.320	<70	[18]
Maximum power point tracking	single chamber	0.300	73	[40]
Maximum power point tracking	single chamber	0.300	67–81	[41]
Integrated circuit-based system	double chamber	0.6–7	<85	[42]
Capacitor-flyback-boost	single chamber	0.300	55	[43]
Flyback-boost converter	chamber	0.44	26	[4]
Integrated circuit-based system and quasi-MPPT	sedimentary	0.236	59.6	This paper

## Data Availability

The data presented in this study are available upon request from the corresponding author.

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
