# Peer review of "Environmental Sensing in High-Altitude Mountain Ecosystems Powered by Sedimentary Microbial Fuel Cells"

_sensors, 2023, doi:10.3390/s23042101_

Round 1
Reviewer 1 Report
In the manuscript entitled “Environmental sensing in high-altitude mountain ecosystems POWERED BY Sedimentary Microbial Fuel Cells” the authors reported to have explored the feasibility of a point-to-point wireless sensors network for environmental parameter monitoring powered by Sedimentary Microbial Fuel Cells (SMFCs). The manuscript has promising results, but low scientific basis. Therefore, I do not recommend it for publication.
Below are some suggestions for improving the manuscript:
1. Authors can make the article more structured/organized. For example: highlights are not included in the abstract; acronyms need to be defined before being used in the text and it is important to have more recent references.
2. Still talking about references, in the Materials and Methods section the authors placed more references than in the Results section. The results need to have more scientific basis/citations.
3. The results (graphs and tables) need to be better explored in the text.
Author Response
Answer to Reviewer 1 (R1) comments and suggestion
The authors thank R1 for his time and thorough review. We are sure that his comments and suggestions improved our report.
R1 comment 1
"Authors can make the article more structured/organized. For example: highlights are not included in the abstract; acronyms need to be defined before being used in the text, and it is important to have more recent references."
Answer 1
Thanks to R1 for his time and careful review; his comments have improved our papers' equality; thanks again. We deleted highlights from the abstract and defined acronyms before being used in the text. We included recent references.
R1 comment 2
"Still talking about references, in the Materials and Methods section the authors placed more references than in the Results section. The results need to have more scientific basis/citations."
Answer 2
Thanks to R1 for his comment. We deleted redundant references from the Material and Methods section. Instead, we gave a more scientific basis by citing recent references.
R1 comment 3
"The results (graphs and tables) need to be better explored in the text "
Answer 3
The authors appreciate R1 suggestion. Therefore, we explore the results more, expand table 6, and cite more recent references in the text.

Reviewer 2 Report
The authors reported “Environmental sensing in high-altitude mountain ecosystems POWERED bY Sedimentary Microbial Fuel Cells” The manuscript well written the following comments ;
1. A list of abbreviation or acronyms must be provided.
2. Pay attention intext Figures and Table must be written as Fig 1 and Table 1
3. Check line 257-261 are bolded needs to be unbold
4. Align the numbering of the equations (1-5)
5. Table 5 ; how was the efficiency defined
Author Response
Answer to Reviewer 2 (R2) comments and suggestion
Thank R2 for his comments and suggestions. R2 review has helped improve our paper.
R2 Comment 1
"A list of abbreviation or acronyms must be provided"
Answer 1
We defined acronyms before being used in the text instead of providing a list of abbreviations.
R2 Comment 2
"Pay attention intext Figures and Table must be written as Fig 1 and Table 1"
Answer 2
We prefer to keep intext Figure and Table instead of Fig and Table as R2 suggests because there is no direct instruction from the Journal.
R2 Comment 3
"Check line 257-261 are bolded needs to be unbold"
Answer 3
We unbold text from L257 to L261
R2 Comment 4
"Align the numbering of the equations (1-5)"
Answer 3
We aligned the numbering of equations 1 to 5.
R2 Comment 5
"Table 5; how was the efficiency defined."
Answer 5
We defied the efficiencies in equations (5) and (6).

Reviewer 3 Report
The main drawback of this work is related with its low scientific soundness. The manuscript presents a case study based on an entire mote design that uses a commercial architecture and highlights the SMF cells and the PMS. The main presented result is that SMF cells "is a feasible technology for remote, on-site moniotoring of physical environmental parameters in high altitude mountanious areas". There is no improvement in the techniques or knowledge regarding SMF cells or new PMS designs or communication protocols that have a direct impact on the energy consumption.
There are also minor corrections as:
Editing problems
Lines 25-30. This paragraph is repeated.
Lines 257-261. bold type is not neccesary here.
Line 112,121,132 ,150 ... Figures, tables and equations in the text appear with yellow markers.
Bibliography: Most of the references have more than ten years old. Recent references are required.
Author Response
Answer to Reviewer 3 (R3) comments and suggestion
Thank R3 for his comments and suggestions.
R3 comment 1
"There is no improvement in the techniques or knowledge regarding SMF cells or new PMS designs or communication protocols that have a direct impact on the energy consumption."
Answer 1
We agree with R3. We do not improve the energy consumption of the power system. However, the scope of this study was to explore the feasibility of a point-to-point wireless sensors network for environmental parameter monitoring powered by sedimentary microbial fuel cells (SMFCs) implemented at 2800 masl. To reach our goal, we implemented a PMS and parallel-connected four SMFCs. We used a fifth SMFC as a reference to implement a quasi-MPPT. And our results show that it is possible to harvest energy from the SMFCs effectively and supply energy to a point-to-point wireless sensor network. The system achieves an overall efficiency of 59.58 %. Furthermore, the control mechanisms prevent energy waste and maintain a stable voltage despite the MFC's high impedance, low time response, and low energy production.
R3 comment 2
"Editing problems"
Answer 2
We edited and proof the whole document. Thanks
R3 comment 3
"Lines 25-30. This paragraph is repeated."
Answer 3
We deleted text from L25 to L30
R4 comment 4
"Lines 257-261. bold type is not neccesary here"
Answer 4
We unbold text from L257 to L261.
R5 comment 5
"Line 112,121,132 ,150 ... Figures, tables and equations in the text appear with yellow markers"
Answer 5
We unhighlighted all the figures, tables, and equations words in the text.
R5 comment 6
"Bibliography: Most of the references have more than ten years old. Recent references are required"
Answer 6
we explored the results more and cited more recent references in the text.

Round 2
Reviewer 1 Report
The authors revised the manuscript and made some improvements, but the results still need to be enriched, better exploring the graphs and tables in the text, making comparisons with what is in the literature or even taking some complementary measure.
Author Response
The authors appreciated the time and thorough review performed by the reviewers. We strongly believe their comments and suggestions have improved our paper's quality.
…………………………………………………………………………………………………………………………………………….
Answer to Reviewer 1 (R1) comments and suggestion
The authors thank R1 for his time and thorough review. We are sure that his comments and suggestions improved our report.
R1 comment 1
The authors revised the manuscript and made some improvements, but the results still need to be enriched, better exploring the graphs and tables in the text, making comparisons with what is in the literature or even taking some complementary measure.
Answer 1
Thanks to R1 for his time and careful review; his comments have improved our papers' equality; thanks again. We gave a more scientific basis by citing recent references. Therefore, we explore the results more, and cite more recent references in the text.

Reviewer 3 Report
Quoting the authors, 'the scope of this study was to explore the feasibility of a point-to-point wireless sensors network for environmental parameter monitoring powered by sedimentary microbial fuel cells (SMFCs) implemented at 2800 masl.' In my opinion, this study has not enough significance content to be published in Sensors.
Author Response
The authors appreciated the time and thorough review performed by the reviewers. We strongly believe their comments and suggestions have improved our paper's quality.
…………………………………………………………………………………………………………………………………………….
Answer to Reviewer 3 (R3) comments and suggestion
The authors thank R3 for his time and thorough review. We are sure that his comments and suggestions improved our report.
R1 comment 1
"Quoting the authors, 'the scope of this study was to explore the feasibility of a point-to-point wireless sensors network for environmental parameter monitoring powered by sedimentary microbial fuel cells (SMFCs) implemented at 2800 masl.' In my opinion, this study has not enough significance content to be published in Sensors."
Answer 1
Thanks to R3 for his time and careful review; his comments have improved our papers' equality; thanks again.
Energy harvesting or scavenging technology has recently gained attention for clean and sustainable energy. This study aimed to explore the feasibility of SMFC operation in the presence of low dissolved oxygen concentrations for remote, on-site monitoring of physical environmental parameters in high-altitude mountainous areas. The implemented power management system (PMS) uses a reference SMFC (SMFCRef) to instrument a quasi-maximum power point tracking (quasi-MPPT) algorithm to harvest energy stably. Instead, the cathode electrodes accept electrons and protons to reduce oxygen to water, taking advantage of the dissolved oxygen gradient of water concerning depth. Hence, the feasibility of SMFC operation under low dissolved oxygen concentration becomes a significant study in high-altitude mountain areas.
Our results show the feasibility of SMFC operation in the presence of low dissolved oxygen concentrations for remote, on-site monitoring of physical environmental parameters in high-altitude mountainous areas.
